

# How Hack distributions of rill networks contribute to nonlinear slope length-soil loss relationships

Tyler H. Doane[1], Jon D. Pelletier[1], and Mary H. Nichols[2]

[1]Department of Geoscience University of Arizona, 1040 E. 4th St. Tucson, AZ, 85720
[2]USDA Agriculture Research Service, 2000 E. Allen Rd., Tucson, AZ, 85719

**Correspondence:** Tyler H. Doane (tdoane@arizona.edu)

**Abstract.** Surface flow on rilled hillslopes tends to produce sediment yields that scale nonlinearly with total hillslope length. The widespread observation lacks a single unifying theory for such a nonlinear relationship. We explore the contribution of rill network geometry to the observed yield-length scaling relationship. Relying on an idealized network geometry, we formally develop probability functions for topological variables of contributing area and rill length. In doing so, we contribute towards a complete probabilistic foundation for the Hack distribution. Using deterministic and empirical functions, we then extend the probability theory to the hydraulic variables that are related to sediment detachment and transport. A Monte Carlo simulation samples hydraulic variables from hillslopes of different lengths to provide estimates of sediment yield. The results of this analysis demonstrate a nonlinear yield-length relationships as a result of the rill network geometry. Theory is supported by numerical modeling wherein surface flow is routed over an idealized numerical surface and a natural one from northern Arizona. Numerical flow routing demonstrates probability functions that resemble the theoretical ones. This work provides a unique application of the Scheidegger network to hillslope settings which, because of their finite lengths, result in unique probability functions. We have addressed sediment yields on rilled slopes and have contributed to an understanding Hack's law from basic probabilistic reasoning.

## 1 Introduction

Rilled hillslopes are common in semiarid, agricultural, and recently disturbed landscapes (Figure 1). In these settings, rills concentrate surface flow and serve as efficient pathways for sediment transport and erosion. There is a long legacy of work that explores the mechanics and consequences of rill processes through field observation, experimentation (Govers, 1992; Liu et al., 2000), and numerical simulation (Hairsine and Rose, 1992; McGuire et al., 2013). This body of work highlights a number of key observations and relationships. Among these is the observation that sediment yield at the base of a hillslope tends to vary nonlinearly with the total length of the hillsope, $q \propto L_h^\beta$ where $q$ [L$^2$ T$^{-1}$] is sediment flux and $L_h$ [L] is the hillslope





length, and $1.4 \leq \beta \leq 2.0$ (McCool et al., 1993; Govers et al., 2007). Here, we consider the role of the rill network geometry in contributing to this nonlinear relationship.

Nonlinear scaling relationships between sediment yield and slope length have been observed on all slopes for which sur-
face flow is a dominant sediment transport mechanism. On unrilled surfaces, the physical reasoning for such relationships calls on the nonlinearities of Manning's equation and flow routing. Moore and Burch (1986) demonstrate that application of Manning's Equation leads to nonlinear relationships between hydraulic variables and slope length. Insofar as sediment detachment varies with hydraulic variables, then the nonlinear relationship extends to sediment yield. That work focuses primarily on unrilled settings, though it briefly addresses the impact of rills which they demonstrate adds nonlinearity for a simple net-
work configuration. Further, the revised universal soil loss equation includes an empirical parameter that explicitly addresses the nonlinear impacts of slope length for rilled hillslopes (Renard, 1997). Curve fitting for the slope-length factor in rUSLE suggests $q \propto L_h^{1.4-1.8}$, where the exponent depends on the rill spacing and the average slope. Here, we present a probabilistic theory that highlights the role of network configuration.

Our probability theory is aimed at developing a formal description of the topological variables of rill flow length, $l$ [L],
and contributing area, $A$ [L$^2$] for an idealized rill network. From this theoretical starting point, we then extend the analysis to hydraulic variables that are related to sediment detachment and transport. This work is related to a suite of previous studies that incorporate probabilistic approaches to rill transport and dynamics. Most notably, our approach is similar to two previous studies. First, Lewis et al. (1994a, b) develop a stochastic model (PRORIL) for rill development and sediment transport that includes variable drainage density and flow rate. In this work, the authors present the model as a tool to explore the development
of rill networks. Second, Damron and Winter (2008) employ a dynamic, but idealized rill network wherein links between nodes can change based on a node's history. They use this model to demonstrate the temporal characteristics of sediment passing by a node as a result of the dynamics that occur in upslope links. Our work here differs as we consider a static and idealized network to develop probability functions from which we sample in a Monte Carlo simulation to provide a robust sample of sediment yields for hillslopes of different lengths.

Other probabilistic approaches have been applied to rill settings. Nearing (1991) consider the probability of particle entrainment as a result of the overlapping distributions of instantaneous shear stress and soil resistance. They demonstrate that this leads to the ability for flows to entrain sediment from soils that are relatively strong. Similarly, Mei et al. (2008) consider the rill width as a random variable, which will influence flow depth and shear stress. Using a linearized perturbation method, they demonstrate the impact on statistical moments of hydraulic variables of flow velocity and depth. Our work considers the
probability involved with the macro-scale patterns of rill networks, and, in principle, could be combined with these efforts that describe dynamics within rills.

We have two goals. First is to provide a rigorous probabilistic description of the rill network. In particular, we wish to formally develop the conditional distribution, $f_A(A|l)$, which is read as the probability distribution of contributing area, $A$, given that a watershed has a length $l$, which is also a random variable with distribution $f_l(l; L)$. These two distributions
combine to create the joint distribution $f_{A,l}(A, l; L)$. This is the Hack distribution which has been extensively studied and used to identify patterns in landscapes, but to date, a complete derivation of the distributions remains to be done (Hack, 1957; Gupta





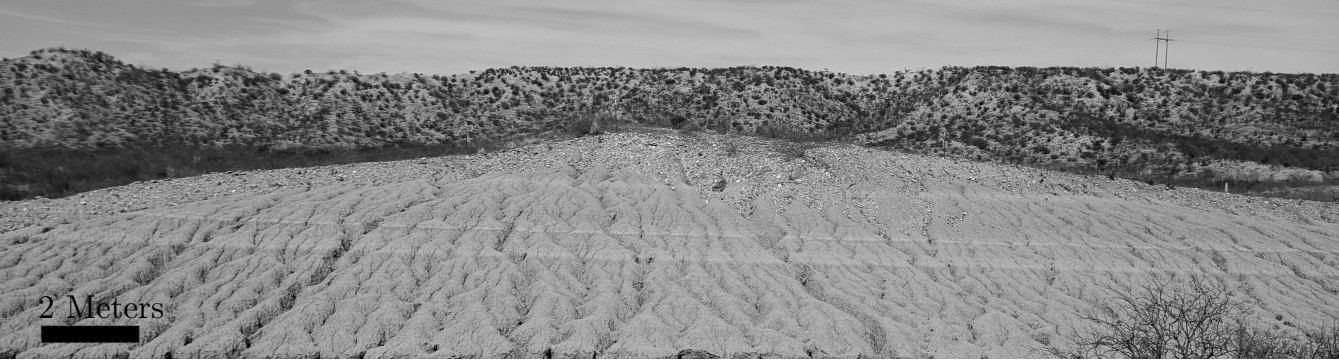

**Figure 1.** A rilled hillslope near Benson, AZ. Prominent sub-horizontal lines are stratigraphy of the lake sediments of the region.

et al., 1996; Dodds and Rothman, 2000). Second, we ask if the particular arrangement of the rills focuses flow such that it leads to a nonlinear sediment yield relationship with hillslope length, $q \propto L_h^\beta$, where $L$ is the total hillslope length. This involves two approaches. First we extend the probability theory for topological variables to hydraulic and sediment transport variables

of unit stream power, shear stress, and sediment concentration. Second, we numerically route flow down the idealized and a natural rill network to evaluate and inform the theory. These results are compared with natural topography from a steep rilled hillslope in northern Arizona.

Before moving on, here is a note about notation. We use $f_x(x; y)$ to denote a probability density or probability mass function for the random variable $x$ with parameter $y$. We use the subscript here to indicate the random variable for the probability

function. This becomes useful later.

## 2 Theory

### 2.1 Network Geometry

We develop a theory for rill network geometry that is based on the Scheidegger model (Scheidegger, 1967). These networks have two characteristics. First, for every unit distance downslope, a rill has equal probability of moving 1/2 unit left or right.

Second, uniform drainage density is maintained, such that where two rills converge, which leaves one downslope node empty, a new rill is generated at the empty node (Figure 2A). These two rules sufficiently describe the network and allow for us to develop theoretical distributions concerning the rill lengths, contributing areas, and flow variables for simple conditions. Other network classes exist including optimal criticality networks (OCN) and Peano basins (Maritan et al., 2002; Yi et al., 2018). Optimal criticality networks are constructed by iterative numerical procedures that minimize the energy expenditure within

the network (Rinaldo et al., 1993). As such, there are a great number of network configurations that satisfy the constraint and there are not clear rules for the construction of links and rill paths. Peano networks are a class of self-similar trees wherein





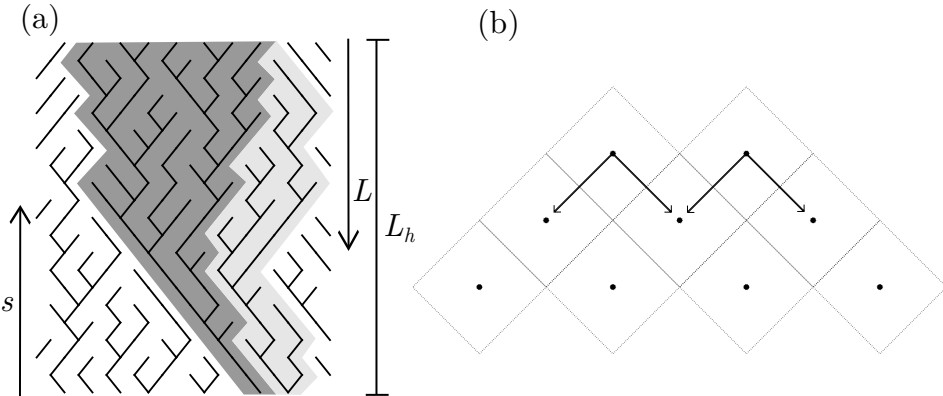

**Figure 2.** (A) Paths of one realization of a Scheidegger network with open (dark gray) and closed (light gray) watersheds highlighted. (B) Illustration of the grid and possible paths of links. Nodes are offset at downslope levels. A square grid is shown here, but there is no requirement that it be square.

perpendicular tributaries are recursively added to the network at finer scales (Gupta et al., 1996). On hillslopes, flow is in one dominant direction, which is not the case for Peano networks so it is unrealistic for our purposes.

The Scheidegger model is highly idealized, but presents a good first-order model for rilled slopes. There is a legacy of using
these networks in geomorphology to explore the statistics and fractal characteristics of typically large-scale river network geometries (Dodds and Rothman, 2000; Maritan et al., 2002). Scheidegger networks are also referred to as discrete webs and their continuous counterpart are known as Brownian webs in other literature. The network in Damron and Winter (2008) is a modified Scheidegger network where the links become dynamic through time.

Central to this work is Hack's Law, which is a nearly universal empirical scaling observation where the mean length of the
main channel for an ensemble of watersheds is related to the contributing area by an exponent,

$$\langle l \rangle = \theta A^m, \tag{1}$$

where $l$ is the length of the main channel, $A$ is the contributing area, $\theta$ is a constant, and the brackets denote an ensemble average. The exponent $m$ is the subject of work that explores the fractal characteristics of networks (Hack, 1957; Dodds and Rothman, 2000; Maritan et al., 2002; Bennett and Liu, 2016). We choose to rewrite Hack's law with $l$ as the independent
variable,

$$\langle A \rangle = \phi l^{1/m}, \tag{2}$$

where $\phi$ is a constant for which $\phi \neq \theta^{1/m}$ (Dodds and Rothman, 2000). We find this form more suitable for the theory developed below. Written this way, (2) is an expression of the mean of the conditional distribution $f_A(A|l)$, the derivation of which is one of our goals.

We begin with an observation on the random walks of watershed divides. Insofar as rills take simple random walks and uniform drainage density is maintained, then watershed divides are also random walks that follow the same rules (Dodds and





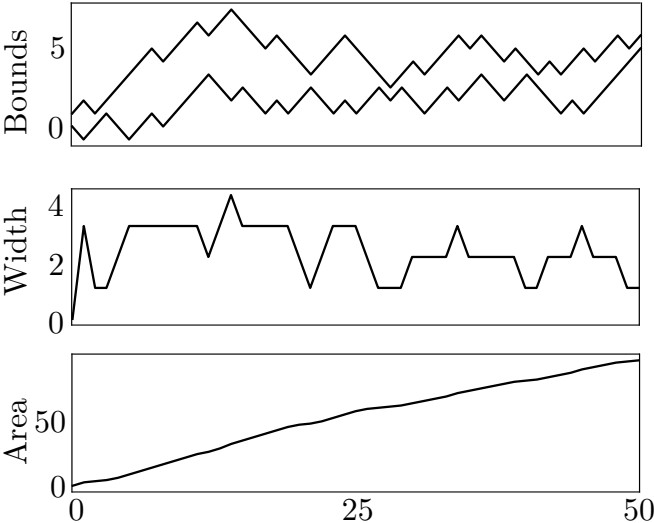

**Figure 3.** Diagrams showing (A) the positions of two random walks that define the boundary of a watershed, (B) the random walk of $w(x)$ and (C) its integral $A(x)$.

Rothman, 2000; Damron and Winter, 2008). The width, $w(s)$ [L] at any particular upslope location $s$ [L], is the difference between two random walks. Characterizing divides in this way allows for the following definitions:

$$w(s) = b_1(s) - b_2(s) \tag{3}$$

$$A(l) = \int_0^l w(s)\,\mathrm{d}s. \tag{4}$$

where $b_n(s)$ [L] denote positions of the two watershed divides, and $w(s)$ is the width function (Figure 3) (Rigon and Ijjasz-Vasquez, 1993; Veneziano et al., 2000; Lashermes and Foufoula-Georgiou, 2007; Ranjbar et al., 2018). The width function for a watershed of length $l$ must always be positive until $w(l) = 0$, indicating the watershed is closed. By necessity, where a new rill is initiated, all probability functions equal unity with zero variance,

$$f_A(A = 1) = f_l(l = 1) = f_w(w = 1) = 1. \tag{5}$$

Moving upwards, the properties of the random walks of $b_n$ completely determine $f(w; s)$ and $f(A|l)$.

The simple random walks of watershed divides move left or right with a distance of $1/2r$ with equal probability, where $r$ is the rill spacing. The width function, being the difference between two of these simple random walks then has three possibilities at each step, $\Delta w = [-r, 0, r]$, which occur with probabilities $P(\Delta w) = [1/4, 1/2, 1/4]$. We recognize $P(\Delta w)$
as the components of the stencil for a central difference solution to linear diffusion (Hornberger et al. (2014)). That is, the probability distribution of $w(s)$ evolves as,

$$\frac{\partial f(w)}{\partial s} = D\frac{\partial^2 f(w)}{\partial w^2}, \tag{6}$$



Earth **Surface**
**Dynamics**
Discussions



where $D = r/2$, initial conditions are given by (5) and boundary conditions satisfy $f(w = 0) = 0$. Note that Eq. (6) is like a Fokker-Planck equation for the width function. The analytic solution of such a diffusion problem (Carslaw and Jaeger, 1959)

is

$$f(w; s) = \frac{2w}{rs} e^{-\frac{w^2}{rs}},\tag{7}$$

which is a Rayleigh distribution. The Rayleigh distribution arises for the problem of the magnitude of the sum of two normally-distributed variates (Siddiqui, 1962). Our problem involves the sum, or difference, of two normally-distributed variates, $b_n$ so this result is consistent with previous work. The moments of a distribution for a random walk are key to understanding the

distribution of its integral, $A(l) = \int_0^l w(s)\mathrm{d}s$. The mean and variance of width from (7) are

$$\mu_w(s) = \frac{\sqrt{\pi rs}}{2}\tag{8}$$

$$\sigma_w^2(s) = \frac{(4-\pi)r}{4}s.\tag{9}$$

For an unrestricted Brownian random walk, (8) and (9) contain all of the information required for the distribution of $A(l)$. In that case $f(A; l) = \mathcal{N}(0, \sigma^2 \frac{l^3}{3})$ (Parzen, 1962), where $\sigma^2$ is the coefficient in (9). Here, however, the requirement that $w(s) > 0$

imparts finite values for the drift, $\mu_A(l)$, changes the scaling between the variance of the random walk and its integral, and introduces finite skewness to the distribution. Because the result has finite skewness, more information would be required to determine the form the distribution. Nonetheless, the first two moments are informative. The mean area involves the integral of $\mu_w$,

$$\mu_A(l) = 2 \int\limits_0^{l/2} \frac{\sqrt{\pi rs}}{2}\mathrm{d}s = \frac{\sqrt{\pi r}}{3\sqrt{2}}l^{3/2},\tag{10}$$

which is a formal expression of Hack's law with $A$ as the dependent variable. We emphasize that this is a complete derivation of Hack's law. Previous work has numerically or empirically demonstrated values of $\phi$ and $m$ (Hack, 1957; Dodds and Rothman, 2000), where $m$ can range from $1/2$, for self similar networks, to $2/3$ for Scheidegger networks (Maritan et al., 2002; Yi et al., 2018). There is little discussion about the value of $\phi$, but it is often determined by fitting distributions or by log-log regression between $l$ and $A$. Equation (10) represents a formal reasoning for both the values of $\phi$ and $m$. Our result is specific

for Scheidegger networks; however, a result like (10) may be obtained if one knows $\mu_w(s)$ and the characteristics of $w(s)$.

We now turn to the variance. From (9) we may obtain $\sigma_a^2(l)$. Once again, if $Z(t)$ is the integral of an unrestricted stochastic process, then $\sigma_Z^2(t) = \sigma^2 t^3/3$ Parzen (1962), but the requirement that $w(s) > 0$ results in a different relationship. We find instead that

$$\sigma_A^2(l) = \frac{\sigma^2}{6} \frac{(l-2)^3}{3},\tag{11}$$

where the term $l-2$ satisfies a requirement that a watershed of length 2 has zero variance for the contributing area. The presence of 6 in the denominator lacks a rigorous explanation; however, we expected that $\sigma_A^2$ increase at a rate slower than what is typical for unrestricted random walks. Placing (9) into (11),

$$\sigma_A^2 = \frac{(4-\pi)r}{72}(l_n - 2)^3.\tag{12}$$



Earth **Surface**
**Dynamics**
Discussions



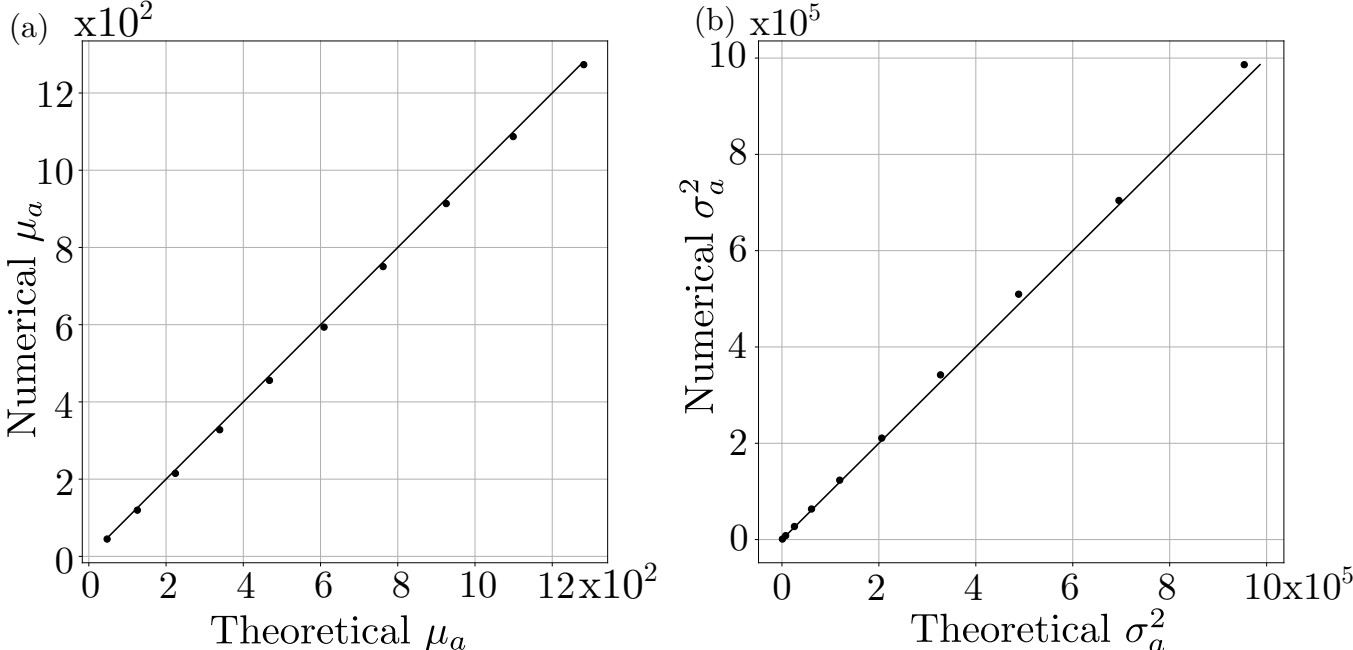

**Figure 4.** Plots of theoretical versus numerical values for $\mu_A(l)$ (A) and $\sigma_A^2(l)$ (B). 1:1 line is shown in black.

There is good agreement between moments from numerical simulations of random walks and theory (Figure 4) and these
moments become parameters of the distribution $f(A|l)$.

Dodds and Rothman (2000) demonstrate that $A(l)$, given a large $l$, is distributed as an inverse Gaussian random variable.
Inverse Gaussian distributions have a foundation in random walk theory where they describe first-passage processes. However,
Dodds and Rothman (2000) state that they identified the Inverse Gaussian as the form by postulating it and fitting parameters.
Here, we rely on their insight but have developed a basis for the moments and therefore have expressions for the parameters
based on the properties of the random walk of $w(s)$. Setting $\alpha = \sqrt{\pi r}/3\sqrt{2}$ and $\lambda = (4-\pi)r/72$ and relaxing the condition
that $\sigma_A^2(l=2) = 0$, the inverse Gaussian distribution is

$$f(A|l) = \sqrt{\frac{\alpha^3}{2\pi\lambda}}\frac{l^{3/4}}{A^{3/2}}e^{-\frac{\alpha(A-\alpha l^{3/2})^2}{2\lambda l^{3/2}A}}\ . \tag{13}$$

As written (19) differs from the result obtained by Dodds and Rothman (2000) for two reasons. First, we use a form of Hack's
law with area as the dependent variable as opposed to length. Second, they have formed a new variable $z = lA^{-2/3}$, where we
have simply kept the distribution as a function of $A$.

We numerically simulate the area enclosed by two random walks 100,000 times for watersheds of length 20 and show that
the form developed here fits numerical distributions better than the form in Dodds and Rothman (2000) (Figure 5). Those
authors limit their analysis to watersheds that involve more than 500 downslope nodes. It is unclear if there should be a
significant difference between large and small watersheds in a Scheidegger model, though we offer it as an explanation for





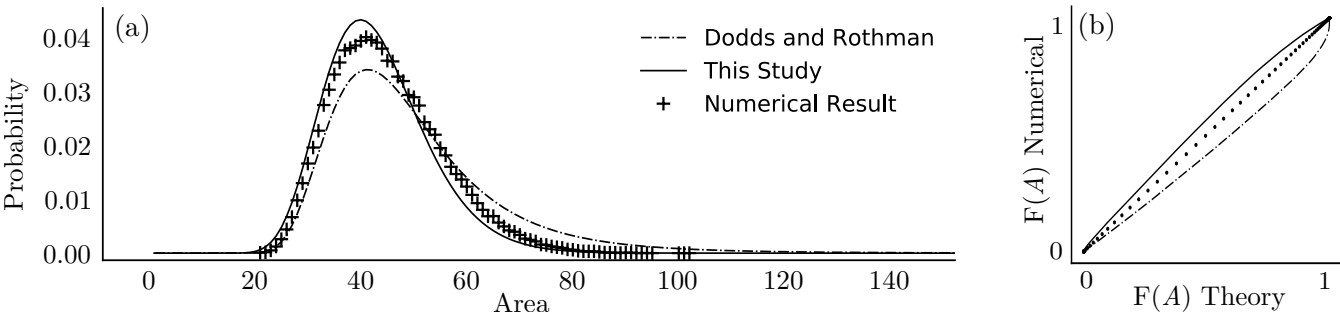

**Figure 5.** A. Probability functions according to *Dodds and Rothman* [2000], this study, and 100,000 numerical simulations of $w(s)$ for which $w(l) = 1$. B. QQ-plot of theory and numerical distributions.

the discrepancy between our study and theirs. Now that we have developed $f_A(A|l)$, we move on to the marginal distribution $f(A; L)$ where $L$ is a distance from the ridge. We rely on

$$f_A(A; L) = \int_1^L f_l(l) f_A(A|l) \mathrm{d}l \,. \tag{14}$$

We now turn to $f(l)$.

Watershed lengths are distributed as a power law (Dodds and Rothman, 2000). We write

$$f(l) = \frac{l^{-3/2}}{2} \,, \tag{15}$$

which is a Pareto distribution with scale parameter 1 and shape parameter $1/2$. $L$, however, is finite and therefore longer watersheds are censored. Though they are censored, the distribution is not simply truncated, but composed of two populations. The first population contains watersheds that have closed within a length $L$. The other population contains the watersheds that are truncated at $L$, but would be longer if $L$ were lareger. Said differently, they are watersheds that would be longer than $L$. Proportions of closed and open watersheds respectively are

$$P(l \leq L) = F(l \leq L) = 1 - L^{-1/2} \tag{16}$$

$$P(l > L) = 1 - F(l \leq L) = L^{-1/2} \tag{17}$$

where $F(l)$ is the cumulative probability function. As hillslopes lengthen, the proportion of open watersheds decays. The complete Hack distribution for a hillslope is a mixture of the two populations and is given by

$$f(A, l; L) = P(l \leq L) f(l) f(A|l \leq L) + \tag{18}$$
$$P(l > L) f(A|l > L) \,.$$

Closed watersheds are addressed with the first term on the right hand side which combines (19), (15), and $F_l(l \leq L)$. Open watersheds are addressed with the second term for which we suggest $f_A(A|l > L)$ is based on the inverse Gaussian, but the



Earth **Surface**
**Dynamics**
Discussions



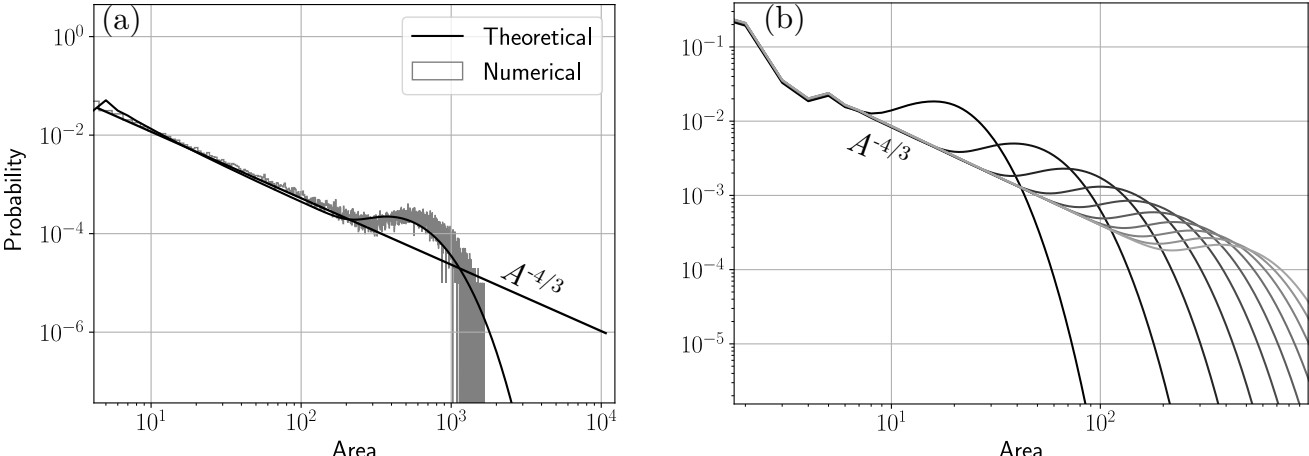

**Figure 6.** (A) Probability distribution of contributing area on a hillslope according to theory and a numerical exercise of 100,000 simulations of $w(s)$ on a hillslope with 100 levels. (B) Probability distribution of contributing area for hillslopes of increasing length.

the variance and mean differs. For this distribution, we suggest $\alpha_o = \sqrt{\pi r}/3$ and $\lambda_o = (4 - \pi)r/12$. A functional form for the complete Hack distribution is

$$
f_{A,l}(A,l;L) = \frac{(1 - L^{-1/2})}{2} l^{-3/2} \sqrt{\frac{\alpha^3}{2\pi\lambda}} \frac{l^{3/4}}{A^{3/2}} e^{-\frac{\alpha(a - \alpha l^{3/2})^2}{2\lambda l^{3/2} A}}
$$
$$
+ L^{-1/2} \sqrt{\frac{\alpha_o^3}{2\pi\lambda_o}} \frac{L^{3/4}}{A^{3/2}} e^{-\frac{\alpha_o(A - \alpha_o L^{3/2})^2}{2\lambda_o L^{3/2} A}} .
$$
(19)

The square root of $L$ grows sufficiently slowly such that the second term is significant on most hillslopes. Our target is the integral of (19) with respect to $l$, for which no analytical solution exists so it must be computed numerically. Numerical integration of (19) reveals an approximate power law distribution, with a notable peak towards the tail which is a result of the second term in (19). A numerical experiment consisting of 100,000 simulations of $w(s)$ for a hillslope with 100 downslope nodes reveals a similar shape to the distribution (Figure 6A). On longer hillslopes, probability is shifted towards the tail (Figure 6B).

The form of $f_A(A; L)$ merits comment. Much of the distribution is characterized by a power law distribution that decays as $A^{-4/3}$, which is a result previously highlighted for large Scheidegger networks (Dodds and Rothman, 2000). This power law relationship results from the first term of (19). However, it is worth noting that even for very long domains, $f_A(A; L)$ will never be entirely monotonic. There will always be some finite probability of a watershed not being closed within that domain. Indeed it is a requirement that at least one watershed be open for a finite domain of any size. When $L$ is very large, this population may defensibly be neglected and $f_A(A) \approx A^{-4/3}$ is appropriate. On hillslopes, this population population is expected to have a significant impact.





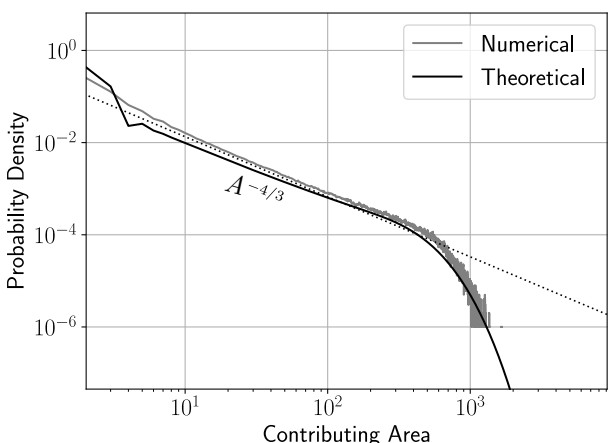

**Figure 7.** Probability density function of total contributing area on a hillslope with length, $L_h = 100$. Dashed line is $A^{-4/3}$.

We emphasize that $f_A(A; L)$ is the distribution of watershed areas at a position that is a distance $L$ from the hilltop. Sediment
detachment; however, occurs throughout the hillslope according to the magnitude of hydraulic variables. The distribution that
informs total hillslope detachment is the complete distribution of contributing area at all points, not just at the terminus of a
watershed. To obtain the this distribution, we introduce, $L_h$ [L], as the complete hillslope length. We reassign an interpretation
of $L$ as a portion of $L_h$ and $L \in [1, L_h]$ (Figure 2A). The distribution is

$$f_A(A; L_h) = \frac{1}{L_h} \sum_{L=1}^{L_h} f_A(A; L).$$
(20)

This states that the distribution of $A$ is the sum of $f_A(A; L)$ for hypothetically short hillslopes, $L$, up to the hillslope length
$L_h$. Numerical computation of (20) produces a monotonically decaying but truncated distribution of the form $f_A(A) \propto A^{-4/3}$
(Figure 7). As $L_h \to \infty$, the truncation disappears. Having demonstrated the form of the distribution of $A$, we now turn to
hydraulic variables.

## 3 Flow Properties

We rely on a set of deterministic relationships to extend the theory for area and rill length to hydraulic variables. For a deter-
ministic, exponential relationship between two variables $x$ and $y$,

$$x = \gamma y^n$$
(21)

and $y$ has a known distribution $f_y(y)$, the distribution of $x$ is

$$f_x(x) = \frac{1}{n\gamma} \left( \frac{x}{\gamma} \right)^{1/n-1} f_y \left[ \left( \frac{x}{\gamma} \right)^{1/n} \right],$$
(22)





and we remind the reader that the subscript refers to the functional form of the distribution for $y$, but the random variable has
changed to $(x/\gamma)^{1/n}$. Using this relationship, we can write probability functions of discharge, rill width, unit stream power, and
shear stress. The task at hand is to generate distributions of these hydraulic variables and perform a Monte Carlo simulation for
sediment detachment on hillslopes of different lengths. First, we must generate the distributions from which we will sample.
We begin by relating area to discharge, $Q$ [L$^3$ T$^{-1}$].

At steady state flow conditions and for uniform runoff, $Q = AR$, where $R$ [L T$^{-1}$] is a runoff rate. Because the relationship
between $A$ and $Q$ is linear, $f_Q(Q; L_h)$ is the same form of $f_A(A; L_h)$. The distribution of discharge is

$$f_Q(Q; L) = \frac{1}{R} f_A\left(\frac{Q}{R}; L_h\right).$$                                (23)

Obtaining the distribution of discharge is key for hydraulic variables that drive sediment detachment.

Previous work addresses sediment detachment in rilled settings [*Hairsine and Rose*, 1992; *Nearing et al.,*, 1991; 1999;
*Giminez et al.*, 2002] which highlights a number of functional forms that relate the volume or mass of detached sediment from
the bed to hydraulic variables. Typically researchers suggest that detachment, $D_s$ [L$^3$ T$^{-1}$] scales as a function of either unit
stream power or shear stress. We first consider stream power.

Unit stream power is a measure of the energy expenditure of surface flow on the stream bed and is written as $\omega = \rho g S h v$,
where $\rho$ [M L$^{-3}$] is fluid density, $g$ [L T$^{-2}$] is acceleration due to gravity, $S$ is fluid surface slope, $h$ [L] is flow depth and $v$
is flow velocity [L T$^{-1}$]. Typical models suggest that sediment detaches as a linear function of $\omega$ (Govers et al., 2007), though
there is evidence that nonlinear relationships exist as well (Nearing et al., 1999). Channel-averaged unit stream power is simple
for rectangular or approximately rectangular channel geometries in which case, $\omega = \rho g S Q / r_w$, where $r_w$ [L] is the rill width.
Therefore, we first must determine the $r_w(A)$ in order to obtain $\omega(A)$.

Previous work demonstrates a relationship between rill width and discharge. Particular values differ between studies, but
in general a relationship $\langle r_w \rangle = kQ^p$ holds where $p$ is a dimensionless exponent that typically ranges from 0.3-0.5 and $k$
[L$^{-2-p}$ T$^p$] is a dimensional coefficient. Gilley et al. (1990) report that $k$ varies over an order of magnitude between $0.2$ and
$5$ depending on the soil type. For simplicity, we set $k = 1$. Torri et al. (2006) present data on rill widths from three different
settings and suggest that the value of $p$ varies from 0.3 to 0.5 for small rills to large gullies. Using this relationship, the unit
stream power is

$$\omega = \frac{\rho g S h v}{w_r} = \frac{\rho g S Q^{1-p}}{k}.$$                                (24)

Rearranging (24) to solve for $Q$ and setting $C = k^{1/(1-p)}(\rho g S)^{1/(p-1)}/R$, we can write the distribution of unit stream power,

$$f(\omega; L_h) = \frac{C}{(1-p)} \omega^{\frac{p}{1-p}} f_A\left(C \omega^{\frac{1}{1-p}}; L\right),$$                                (25)

where again, $f_A(x; L_h)$ refers to $f_A(A; L_h)$ where the random variable $A$ has been replaced with $x$. To obtain this result
requires integrating over $l$ as is the case for area because there is no analytical solution to the integral (Figure 8a). The general
form of the distribution is similar to $f_A(A; L)$, though it decays at a different rate which depends on the value of $p$. The power
law portion of the distribution decays as $\approx \omega^{-3/2}$ for $p = 1/3$.





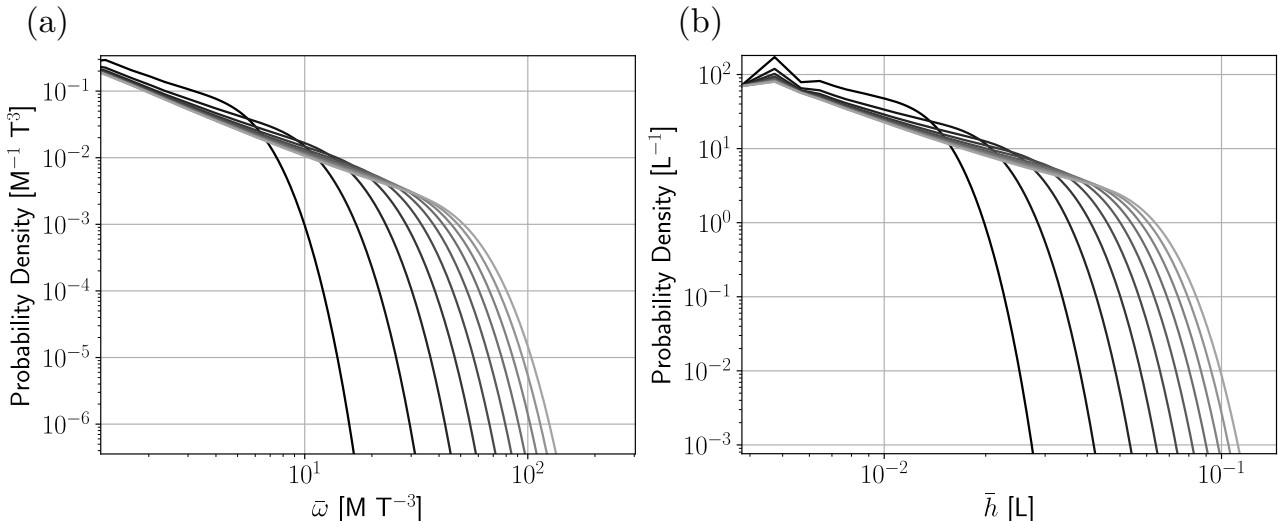

**Figure 8.** Probability density functions for (A) $\omega$ and (B) $\bar{h}$ for hillslopes with $L_h \in (10, 200)$ by increments of 10. Longer hillslopes are lighter colors. The distribution is not smooth for small values of $\bar{h}$ because of the discrete calculation.

Unit shear stress is another hydraulic variable that is often related to sediment detachment rates (Nearing et al., 1999; Govers et al., 2007). Shear stress is written as $\tau = \rho g S h = \omega/v$. Both $h$ and $v$ are unknown, but are related by Manning's Equation,

$$v = \frac{r_h^{2/3} S^{1/2}}{n}, \tag{26}$$

where $n$ is Manning's roughness coefficient, and $r_h = w_r h/(2h + w_r)$ is the hydraulic radius. For our planar hillslope, $S$ is uniform so we only need to solve for $h$. Setting $v = Q/w_r h$ we can solve for $h$,

$$h = \left(\frac{4}{5}\right)^{2/3} \frac{n}{S^{1/2}} \frac{(aR)^{1-5/3m}}{k^{5/3}}. \tag{27}$$

Following similar steps for $f_\omega(\omega; L)$ we are able to write out $f_h(h, l)$ which must be numerically integrated to obtain $f(\omega; L_h)$ (Figure 8b). Again, the distribution is a truncated power law that decays as $\bar{h}^{-7/4}$ when $p = 1/3$.

A third detachment model involves the concept of transport capacity, wherein the flow accumulates sediment at rates that are inversely proportional to the concentration (Lewis et al., 1994a; Polyakov and Nearing, 2003). As a flow increasingly entrains more sediment downslope, the sediment concentration in the flow asymptotically approaches a maximum value. As typically written, transport capacity is a geometric variable, not a hydraulic one. A common conceptualization is (Polyakov and Nearing, 2003),

$$\frac{dc}{dx} = \kappa \left(1 - c/T_c\right), \tag{28}$$

where $T_c$ is a maximum concentration that a flow can sustain, $\kappa$ [L$^{-1}$] is an empirical coefficient, $c$ is concentration, and $x$ is downstream distance. Here, we use a volumetric form of concentration so $c$ is dimensionless. We replace $x$ with $A$ and solve





for $c$,

$$c(A) = T_c \left( 1 - e^{-\frac{\kappa}{T_c} A} \right), \tag{29}$$

which may be rearranged to make $A(c)$ such that we may obtain $f_A(c; L)$ as we have done for $\tau$ and $\omega$.

We numerically generate samples of $\omega$, $\tau$, and $c$ by inverse transform sampling from $f_A(A; L)$ and applying the deterministic relationships laid out above. Inverse transform sampling is a method that may be employed to randomly sample from any probability distribution. The method first generates a random sample of values from a uniform distribution between zero and one. The random sample is then translated to values of the random variables (in this case $A$) by mapping the values of the

random sample to those of the cumulative distribution function which ranges from zero to one as well. This is equivalent to sampling from $f_A(A; L)$, but allows for us to do so for any distribution - even those that must be numerically integrated as is the case here.

Inverse transform sampling provides samples of $A$ and equations (25), (27), and (29) translate it to a sample of hydraulic variables. We consider hillslopes of lengths $L_h$ and with $N$ rills at the first level ($s = 0$). To generate samples for an entire

hillslope requires $NL$ samples from $f_A(A; L)$, which corresponds to $NL_h$ nodes. For each node, we obtain a sample of unit shear stress and stream power. Between nodes, rills accumulate flow in a linear fashion and we use the average values of $\tau$ and $\omega$ within a single link. The volume of detachment, $D_s$ [L$^3$ T$^{-1}$] within a link is the area of the channel bed in the link multiplied by the detachment relation,

$$D_s \propto y^\eta w_r \Delta l, \tag{30}$$

where $y$ is a placeholder variable for $\tau$ and $\omega$, and $\eta$ is an exponent. We then take the sum of all detached sediment over the entire hillslope. Assuming a detachment limited system and no deposition, then the cumulative detachment divided by the slope width comprises the sediment flux at the base of the hillslope.

Sampling for sediment concentration requires a slightly different procedure. Sediment concentration at any given point is the cumulative result of all upslope detachment. Therefore, we only need to know $c$ at the base of the hillslope, we sample

from $f_A(A; L_h)$ $N$ times to obtain samples of $Q_s(L_h) = Qc(L_h)$ where $Q_s$ [L$^3$ T$^{-1}$] is the volumetric sediment discharge. The flux is $q = Q_s/Nr$.

Results from the Monte Carlo simulation demonstrate nonlinear relationships between hillslope length and cumulative sediment discharge at the base (Figure 9A). As the power relationship between detachment and hydraulic variables increases, so too does the exponent that relates hillslope length to cumulative sediment flux (Figure 9B). The observed range of the power

relationship places $1.4 \leq \beta \leq 1.9$ and many of our simulations fall within that range. Nearly all simulations that use $c$ with different rate constants fall within the observed range. Detachment models involving $\tau$ and $\omega$ tend to result in flux-length relationships that are too strongly nonlinear. Our assumption; however, that all detached sediment exit the system is likely a simplification. If deposition were included in this model, it would reduce the nonlinear relationships possibly to near or within the observed range.

The sampling method highlights an interesting sidebar. The theory developed above is for highly idealized networks. There are strict requirements for drainage density, flow directions, rill width, and hillslope shape (rectangular). Under strict conditions,





the sum of contributing area at the base of a hillslope must equal the total area of the hillslope. For a hillslope with total width $Nr$, $N$ samples from $f_A(A; L)$ should sum $LNr$. Such a result only occurs with very small probability and more often the sample hillslope area is only approximately $LNr$. This implies that one or some of our strict requirements have been relaxed.

That is, our samples might represent a hillslope that is not entirely rectangular, or where drainage density is not exactly maintained. Such an outcome is a direct result of Monte Carlo simulations and is not novel, but this system highlights the fact that a sampling from an idealized distribution yields a sampled system that is not idealized.

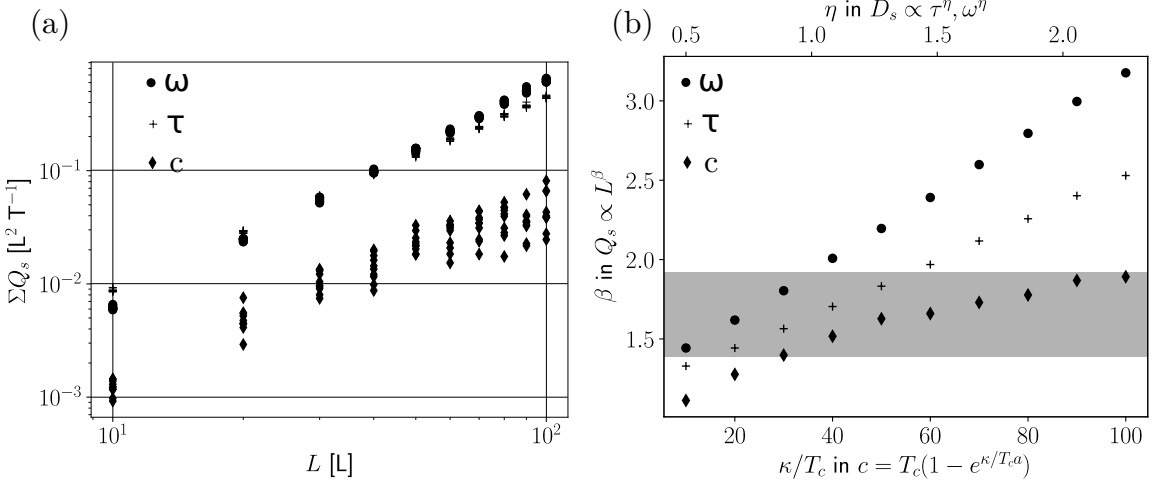

**Figure 9.** (A) Cumulative volume of detached sediment on hillslopes of length $L$ calculated by unit stream power, shear stress, and sediment concentration when $w_r = kQ^{1/3}$. (B) Best fit power-law relationships for different sediment detachment rules (top axis) and rate constants, $\kappa/T_c$, for sediment concentration. The range of observed nonlinear relationships are higlighted in gray.

## 4 Numerical Modelling

We demonstrate these distributions with a simplified numerical model that [1] generates topography with a Scheidegger net-

work of rills and [2] simulates the steady state overland flow using Manning's equation and a numerical flow routing procedure (Pelletier et al., 2005). We simulate steady-state overland flow for a couple of reasons. First, our goal is to demonstrate how the variance of hydraulic variables increases with hillslope length. Steady state flow conditions accomplish this task. Second, numerical simulations show that, depending on the slope, runoff variables rapidly approach steady state values within the first 20 minutes of heavy rainfall and change slowly afterwards (Liu and Singh, 2004). Last, part our goal is to illustrate a first-order

behavior and the details of the hydrograph are not considered here.

To generate topography, the numerical model develops a mask of cells that identify the location of rills that satisfy the two rules of Scheidegger networks. Topography is then generated by imposing some uniform lowering rate within the rills and performing linear diffusion on the interrill areas. This leads to approximately parabolic topography in interrill areas. For the




Earth **Surface**
**Dynamics**
Discussions

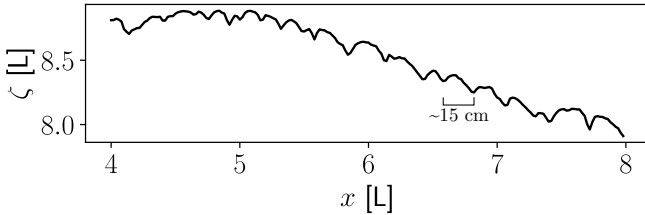

**Figure 10.** Profile of a section of the natural hillslope highlighting relatively uniform rill spacing.

theory developed above, we assume rectangular channels so that flow depth is distributed evenly across-channel. In order to
best match theory to the condition for numerical modeling, we enforce a rectangular channel of uniform width. Under this
condition, the distribution of discharge will remain the same as theory, but hydraulic variables will differ because they depend
on channel width. However, $w_r$ is a function of $Q$ and so the numerics can be mapped to theory.

The natural hillslope is from a steep slope in northern Arizona, in the badlands topography of the Painted Desert. The
hillslope was scanned using a high resolution terrestrial lidar scanner, which yielded topography at 2 cm spatial resolution. The
average slope is 1.3 and rill spacing is relatively uniform at about 15 centimeters (Figure 10). The slope is sufficiently steep
that we anticipate this particular hillslope is detachment limited.

The numerical modeling routine routes flow using a D-infinity scheme combined with Manning's equation to simulate steady
state conditions. The model iteratively applies a uniform rate of runoff to the surface which is routed downslope according to
D-Infinity. For each iteration, Manning's equation solves for depth assuming that it approximates the hydraulic radius (Pelletier,
2008). After each iteration, the depth is updated accordingly and the routine repeats until it approaches a solution to a steady
state configuration of flow depth. This workflow continues until either a threshold of change in depth is reached or a set
number of iterations occur. For this work, the threshold for change in average depth between any two iterations is 1%, or about
50 iterations for these hillslopes.

Routing flow down the idealized and natural surfaces reveal steady state patterns of hydraulic variables (Figure 11 and Figure
325   12). Probability distributions from the simulated surface support the theory developed above. The distributions of contributing
area and discharge reflect the form of (20) and (23) (Figure 13). The distribution of $\omega$ is a deterministic function of $Q$, and so
the distribution is not shown. Furthermore, because we have specified that our idealized hillslope has uniform slope, $h$ is the
only variable in $\tau$ that can change and so we plot the distribution of $h$.

Plots of exceedance probabilities for $A$, $Q$, and $h$ (Figure 13) from the idealized surface reveal similar patterns to theoretical
330   distributions (Figure 8). As hillslopes length, or we sample to progressively lower parts of the hillslope, probability is added to
the tail of all empirical distributions. There is good agreement between distributions of geometric variables ($A$ and $Q$) between
the idealized case and the natural one (Figure 13A and B), which suggests that our theory accurately describes the arrangement
of rills. This lends confidence to our Monte Carlo simulation and the implications for the scaling between hillslope length and
sediment flux.



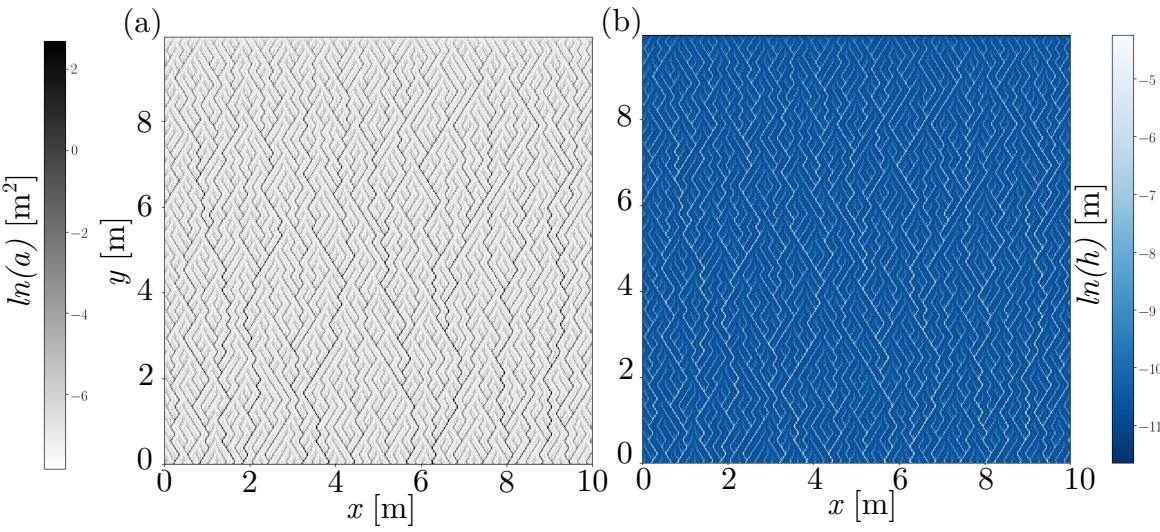

**Figure 11.** (A) Map of contributing area of half of an idealized hillslope. Color scale is in log scale to make small rills visible and highlight the entire network. (B) Map of steady state flow depths according to our numerical model. Color scale is in log scale.

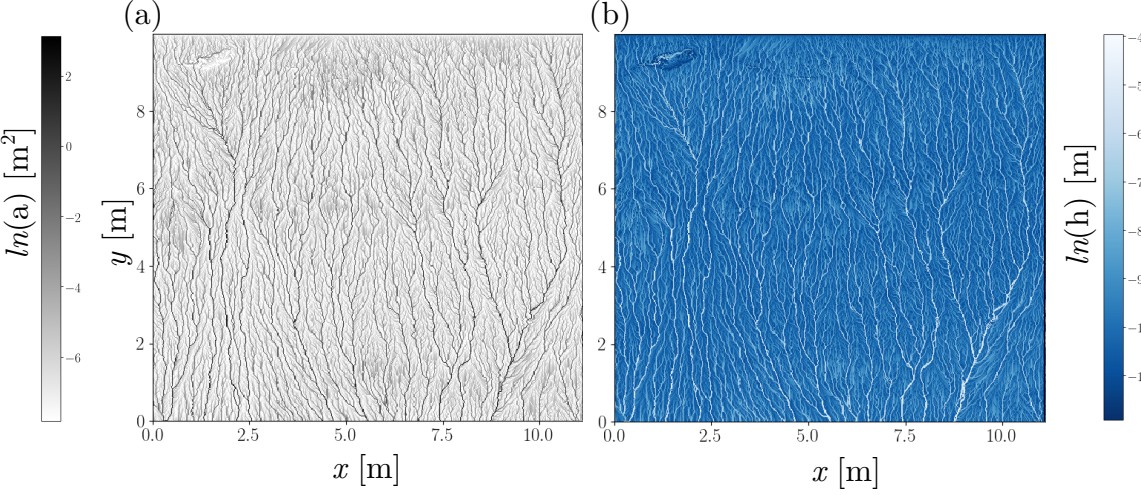

**Figure 12.** (A) Map of contributing area on half of the hillslope. Color scale is in log scale to make small rills visible and highlight the entire network. (B) Map of steady state flow depths according to our numerical model. Color scale is in log scale.



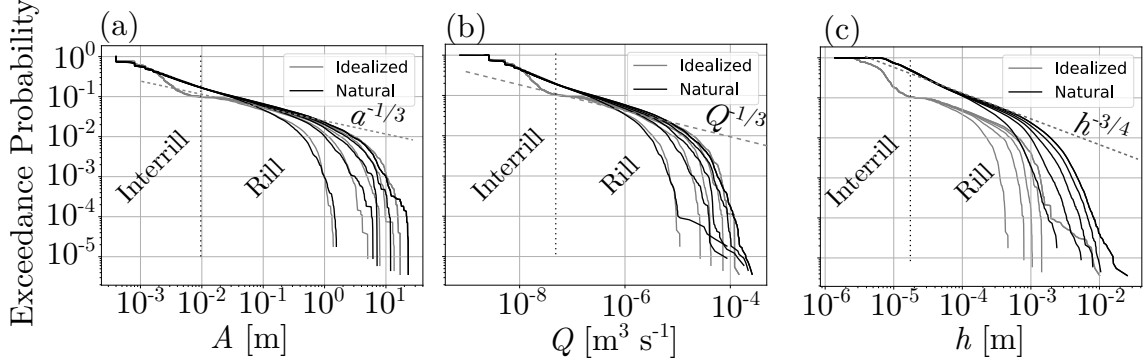

**Figure 13.** Exceedance probability plots for (A) contributing area, (B) discharge, and (C) flow depth from the hillslope in northern Arizona. Log-log slopes from theory are plotted on top of data. The log-log slope in (C) is for $p = 1/3$ in $r_w = kQ^p$.

335    Though geometric variables of $A$ and $Q$ match well, there is notable difference between natural and idealized distributions of $h$. The forms are again similar; however, the location of truncation for the idealized case is about half an order of magnitude shallower than that for the natural hillslope. There are two reasons for this discrepancy. First, for the idealized case, rill widths are uniform. Second, the natural hillslope is rough and the bed slope contains some noise. Therefore, reductions in slope or the quasi-random narrowing and widening of channels drives an increase in flow depth (Mei et al., 2008). Uncertainty in the 340    spatial patterns of channel width have a significant impact on the distributions. Coupling the work here with a more detailed treatment of $w_r$ may yield interesting results.

## 5    Discussion

We have contributed to a formal development of the probability functions for topological variables of $A$ and $l$ for the Scheidegger model. The mathematical steps involve (1) recognizing the width function as a Brownian random walk, (2) developing the 345    Fokker-Planck-like equation for $f_w(w)$, (3) calculating statistical moments for $A$ based on those for $w$. These steps should be appropriate for many networks; however, they are most applicable to Scheidegger-like networks. By this, we mean networks for which there is a single obvious downslope direction and the surface is roughly planar such that channels do indeed take random walks. This is the case for hillslopes, channels on alluvial fans (McGuire and Pelletier, 2016), and perhaps some large-scale river networks. It is clear that if one can characterize the paths of divide lines as some one-dimensional random walk, 350    then the contributing area becomes the integrated random walk and the steps above hold. Scheidegger networks are simply a special case where the divide lines and the channels are probabilistically the same. This may not be true for other networks.

Though we have developed the moments of the distribution of $A$, some items remain outstanding. First, as we have noted the variance increases at a rate six times slower than that of an unrestricted integrated random walk. We suggest this arises from the requirement that the random walk always be positive. However, we currently lack a theoretical explanation for the presence 355    the value of six. Further, we have relied on the work of *Dodds and Rothman* (2000) for the form of the distribution. Although





the Inverse Gaussian distribution has its foundation in random walk theory, the formal development of the distribution from considering the properties of the random walks remains to be done. We anticipate that the demonstration of $f_w(w; s)$ can contribute towards this, because, in principle the distribution a random walk is related to that of its integral.

The theory that we have developed is intended to capture the essence runoff-driven entrainment. However, it does not
consider all processes of entrainment, namely the role of rainfall detachment [*Hairsine and Rose,* 1992, *McGuire et al.,* 2000]. We have not included a theoretical treatment of this process though it may serve to reduce the nonlinear sediment yield-length relationship. The role of raindrop impacts is greatest on bare surfaces and declines as flow depth increase. In our rilled settings then, rain drop detachment will be greatest at the top of the hill and will decline downslope. This is the opposite trend that we see for flow-driven detachment, which only increases downslope. If one were to incorporate raindrop detachment into the
theory developed above, it would tend to reduce the nonlinear relationship between sediment yield and hillslope length. We note that Figure 9 shows nonlinear relationships that are stronger than we typically observe. Therefore, including raindrop impact may contribute to more reasonable scaling relationships. To be clear, this only impacts sediment yield calculated from $\omega$ or $\tau$ and not concentration, which implicitly incorporates all detachment processes and deposition.

Numerical flow routing highlights the success and challenges of applying the theory developed above. To first order, the
arrangement of rills in a network like the idealized one we have used here describes the flow routing on natural hillslopes. This is evident from the distributions of contributing area and discharge. Results shown in Figure 13A and B highlight that for both cases, these distributions decay as power law distributions with exponents close to the theoretical $-4/3$. There are; however, distinct differences between them. First, we note that in the idealized case, exceedance probabilities $R_A(A; L_h)$ and $R_Q(Q; L_h)$ appear to decay faster than the $1/3$ predicted from theory. This may indicate that the idealized Scheidegger model
may not be a perfect description for this network. As mentioned above, other network classes exist such as OCN; however, those networks lack clear probabilistic rules and therefore make developing a theory challenging. We emphasize that despite the slight difference in power-law relationships, the distributions are truncated at remarkably similar locations which leads to similar scaling relationships.

Another difference is apparent in the distinction between interrill and rill contributions to the distributions. For the idealized
case, the distinction between rills and interrills is clear where the interrill portion of the distribution is distinctly not a power law. The same distinction is not clear in the natural slope. We hypothesize that interrill and rill portions do not separate clearly because of the rough topography in the natural hillslope which, even in the interrill areas, tends to focus flow to some degree. The idealized hillslope lacks all roughness so that there is no variance in flow for the interrill areas.

We have specified that the mean channel width increase nonlinearly as $\langle r_w \rangle = kQ^p$. For the case where $p = 1/3$, we expect
$R_h(h; L_h)$ to be a truncated power law that decays as $-3/4$ for our idealized case. Indeed, this is the slope of exceedance probability for the natural slope shown in Figure 13C despite slight differences in the geometry. The shape of $R_h(h; L_h)$ depends on $p$ and the shape of $R_A(A; L_h)$. Assuming that the deterministic relationships hold, we can solve for $p$ given the slopes of the power law portions of $R_A(A; L_h)$ and $R_h(h; L_h)$. Doing so, we find $1/5 < p < 3/10$ for the natural case, which represents the lower range of values from Torri et al. (2006).



Probability theory is the foundation for work that describes the behavior of a cohort of particles (Martin et al., 2012; Fathel et al., 2016; Wu et al., 2019; Pierce and Hassan, 2020) that begin their motions at a common location and time. Also referred to as tracer problems, research in this area often targets how that cohort of particles disperses through time. The majority of this work is with regard to transport in fluvial systems where particles take a great number of hops and intervening rest times over timescales that are appropriate for human observation. On hillslopes, particle motion is infrequent and observation of a great number of individual motions involving a cohort of particles is not practical for most settings. Rilled hillslopes; however, offer a unique setting where particles may move frequently. Though an empirical or experimental component of this work remains to be done, Lisle et al. (1998) present probability theory that informs particle dispersion for a rilled setting. However, they consider a single rill which may or may not nonlinearly accumulate flow in the downslope direction. We have demonstrated a probabilistic framework for the rate of flow accumulation downslope, and, in principle, could be used as a basis for further work exploring particle dispersion or residence times on rilled slopes.

## 6 Conclusions

We have demonstrated probability functions of geometric and hydraulic variables for rilled hillslopes. The theory represents an application of Hack's Law and Hack statistics to hillslopes. The limited space of hillslopes introduces a fundamental difference from the typical application of network scaling arguments (Dodds and Rothman, 2000). We show that the arrangement of rills can lead to nonlinear relationships with sediment detachment which are similar $Q_s \propto L_h^{3/2}$ that is typically observed in nature (Moore and Burch, 1986; Liu et al., 2000; Govers et al., 2007). Flow routing numerical simulations on idealized and a natural hillslope demonstrate agreement between geometric probability distributions - lending merit to the theory.

In pursuing a theoretical form for the distribution of hydraulic variables on hillslopes, we have developed formal expressions for the probability functions of geometric variables. Notably, from considering the properties of random walks that define drainage areas, we have developed the joint probability function of area and length. Building on the work in Dodds and Rothman (2000), we have provided a probabilistic basis for the moments of the conditional distribution, $f(A|l)$. The first moment of this distribution is the well known Hack's Law. This result is specific to Scheidegger networks, but the mathematical steps extend to others.

The work presented above is a combination of probability and determinism. We have relied on simple, but demonstrated deterministic relationships to extend our understanding of the geometry to hydraulic variables. This represents an attempt to explain the first-order behavior. The theory provides a foundation to consider more detailed and stochastic elements of rill networks such as channel geometry and width variations, variable slope, and the consequences of storm-driven hydrographs.



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





| Symbol | Variable | Units |
|---|---|---|
| $\alpha$ | Constant | $L^{1/2}$ |
| $A$ | Contributing Area | $L^2$ |
| $b$ | Lateral position of divide line | $L$ |
| $\beta$ | sed. yield - length exponent | $-$ |
| $c$ | Sediment Concentration | $-$ |
| $D$ | Probability Diffusivity | $L^2$ |
| $D_s$ | Sediment detachment | $L^3 T^{-1}$ |
| $g$ | Acceleration due to gravity | $LT^{-2}$ |
| $\gamma$ | Placeholder coefficient | $-$ |
| $h$ | Flow depth | $L$ |
| $k$ | Discharge-rill width coefficient | $L^{-p-2}T^{-p}$ |
| $\kappa$ | Sediment concentration coefficient | $L^{-2}$ |
| $l$ | Channel length of closed watershed | $L$ |
| $L$ | Downslope distance from ridge | $L$ |
| $L_h$ | Total Hillslope Length | $L$ |
| $\lambda$ | Constant | $L$ |
| $m$ | Hack Exponent | $-$ |
| $\mu_x$ | Mean of variable $x$ | Units of $x$ |
| $n$ | Manning's coefficient | $L^{1/3}T$ |
| $\eta$ | Placeholder exponent for detachment models | units vary |
| $\phi$ | Hack Coefficient | $L^{2-1/m}$ |
| $\rho$ | Fluid density | $ML^3$ |
| $p$ | Discharge-rill width exponent | $-$ |
| $q_s$ | sed. flux | $L^2 T^{-1}$ |
| $Q$ | Water discharge | $L^3 T^{-1}$ |
| $R$ | Runoff | $LT^{-1}$ |
| $R_x$ | Exceedance Probability for random variable $x$ | $-$ |
| $r_h$ | Hydraulic Radius | $L$ |
| $r$ | Interrill spacing | $L$ |
| $r_w$ | Rill width | $L$ |
| $s$ | Upslope distance | $L$ |
| $S$ | Fluid surface slope | $-$ |
| $\sigma_x^2$ | Variance of variable $x$ | units of $x^2$ |
| $\theta$ | Hack Coefficient | $L^{-m+1}$ |
| $\tau$ | Shear Stress | $ML^{-1}T^{-3}$ |
| $T_c$ | Maximum sediment concentration | $-$ |
| $v$ | Depth-averaged flow velocity | $LT^{-1}$ |
| $\omega$ | Stream power | $ML^{-3}$ |
| $w$ | Watershed width | $L$ |