# Peer review of "How Hack distributions of rill networks contribute to nonlinear slope length-soil loss relationships"

_Earth Surface Dynamics, 2020_

## Referee Comment (RC1) · Anonymous Referee #1 · 11 Oct 2020

In this generally well-written manuscript, the authors present a simple probabilistic argument for explaining the Hack distribution and the relationship between sediment yield and hillslope length. The research is sound and interesting, and the case study lends support to the theory. The limitations of the study, largely arising from the assumption of a Scheidegger network, are clearly explained in the discussion. There are however a few logical and mathematical steps that are needed to substantiate the theory.

First, my main concern is with equation (2), from which many of the main results on fA follow. It is not clear how the authors obtained it and what assumptions are involved. In principle, the ensemble average of A, given I, is  $\langle A \rangle = \int A(f_l(l|A)f_A(A))/(f_l(l))dA$ .

So, the ensemble average of A should contain information about the entire joint probability distribution. How does it reduce to equation (2)? Is  $\varphi$  really a constant?

Second, it should be better emphasized what aspects of the theory are new. From the abstract and the introduction, it seems the authors are going to develop a complete and unifying theory for the Hack law and distribution. The work however builds a lot on previous work (e.g., Dodds and Rothman) and there are many assumptions along the way, so it is hard to keep track of what is new and what is merely a review.

Minor comments.

I strongly suggest dividing section 2.1 in subsections. Here the theory is long and very involved. I believe that with subsections, it could be easier to identify and connect the different pieces of the theory as well as recognize the key new aspects. This also applies to section 3.

Line 150. This is the first time  $\alpha$  and  $\lambda$  appears, without being introduced.

Line 153. I think the authors meant (13) rather than (19). Same in line 176.

Line 169. Correct "Larger".

Line 304. Missing "of" in "last part of our goal".

**ESurfD**

---

## Referee Comment (RC2) · Anonymous Referee #2 · 19 Oct 2020

article [utf8]inputenc

**ESurfD**

Interactive
comment

**Comments on "How Hack distributions of rill networks contribute to nonlinear slope length–soil loss relationships"**

October 19, 2020

The manuscript discusses results from the Scheidegger model in terms of the distribution of drainage area and the power scaling of drainage area and length (i.e., Hack's law). In general, I found the paper interesting; however, I have some concerns regarding the methods and derivation of the results. The main problem tackled in this work can be re-stated as finding the distribution of the area under a Brownian motion trajectory before its first return. However, I am not convinced-given the explanation in the current version-that this has been achieved. Especially the solution in Eq. (7), which is the basis of the majority of the results, is very problematic and needs a lot more explanation. In what follows, I try to highlight my main concerns about the presented derivation, followed by some general and minor comments.

Starting from Eq (5), this equation does not read as it meant "a unity probability with zero variance". It should be expressed as an atom of probability at $w = 1$ and zero for other w. Second, should it be $w = 0$ instead of $w = 1$? Right above this paragraph, it is stated that watersheds are closed at $w = 0$; thus they should also initiate at $w = 0$.
The transition to Eq (6) is somehow abrupt. Up to this point, the paper paints a discrete picture of the rills network with r being the scale. Now we see a diffusion equation that is indeed defined in a continuous domain. Does this mean $r \to 0$ is assumed here? If so, the initial and boundary conditions should take the form of the Dirac Delta. Perhaps write the stochastic equation for w and then Eq (6), which is the corresponding FP. Also, keep the s dependence of function f just to makes things clear. Eq (7) should be written as $f(w, s)$; s is a variable –similar to time- not a parameter. Otherwise, it becomes very confusing when the authors talk about initial conditions.

Line 113: It is not clear where this boundary condition comes from? Is this meant to model the closing (termination) of watersheds at $w = 0$. If so, I do not think this is valid. If we think of $w(s)$ as random trajectories which follow a Brownian motion and die out when $w(s) = 0$, there should be an atom of probability at $w = 0$ for each s (rather than zero probability) to model all trajectories that die out at that s. In line 123 it is stated that for "For an unrestricted Brownian random walk ..." which makes me believe this section, including Eq 7, considers an unrestricted stochastic process. In that case, we have a diffusion equation,

$$\frac{\partial f(w, s)}{\partial s} = D \frac{\partial^2 f(w, s)}{\partial w^2}, f(w, s = 0) = \delta(w) \tag{1}$$

This equation gives a solution in the form of normal distribution rather than Rayleigh. This should be clearer. Line 117- "The Rayleigh distribution arises for the problem of the magnitude of the sum of two normally distributed variates". This statement is very confusing. The authors wrote Eq (7) for w, which indicates w is a Brownian motion and has no information whether it is a sum of two variables. Basically, w is a Brownian motion itself, and there is no reason it behaves differently than its constituents. Although the difference may come from the boundary conditions for which I expressed my concern earlier.

At some point, I got the impression that the authors meant to fix length "l" and basically

look at all trajectories of w, which starts at $w(s = 0) = 0$ and end at $w(s = l) = 0$. If this the intention, I am not sure if the linear diffusion still holds for those trajectories as we are sampling the trajectories in a very specific way.

Some parts of the paper give the readers the impression that the Scheidegger model is in the same class as, for instance, OCNs. This is not true and very misleading. OCN describes the steady-state solution of a sediment balance and satisfies both the continuity equation of water and sediment. However, the Scheidegger model only heuristically gives "a solution" for the water continuity equation in the form of a network that may not be achievable from any flux-based model.

Eq 11. It is not clear how this is achieved. Is it an empirical equation?

Fig 2- The label of the x-axis is missing.

Line 17- "Efficient" in what sense? Energy? Flux? How do we know they are efficient?

Rill flow length $l$ needs to be defined in the intro.

Line 55- The second question is not clear! Try to rephrase.

Line 73- Optimal Channel Network (OCN)?

Line 75- "As such, . . .": Is this referring to OCN? What is the "constraint"? OCN has a "clear rule" to construct networks.

---

## Referee Comment (RC3) · Anonymous Referee #3 · 24 Oct 2020

[11pt]article sectsty amsmath

**Comments on:**
**How Hack distributions of rill networks contribute to nonlinear slope length-soil loss relationships**

[Figure]

**1 Review Recommendation**

Overall, the paper is of interest and, to a significant degree, addresses the two main goals of (1) exploring the non-linearity of the slope length-sediment output relation and (2) deriving Hack's law for Scheidegger networks. The main reason that this reviewer found the results of the analysis to be convincing lies in the excellent comparisons between the theoretical results and the numerical results from the simulations, as illustrated in figures 4, 5, 6, and 7. While these figures probably reflect an appropriate theoretical background, it is possible that they reflect the robustness of the modelling approach.

The paper as it stands could be significantly improved with some rewriting and, as such, would be quite publishable. Since the primary goal is to demonstrate and understand the non-linearity of the length-sediment relation for a rilled surface, it would have been useful to have first discussed the relation for an unrilled surface. There are clearly reasonable sediment transport laws that lead to a non-linear response in such a case, especially since the existence of rilling indicates that the flow regime is characterized by erosional instabilities. It would then be important to attempt to explain any differences between the planar and rilled flows.

It would be useful to make a little section for related literature, and especially to add a comprehensive summary of all of the relevatn findings of Dodds and Rothman since they are employed/referred to so extensively.

The section on network geometry could be greatly improved for ease of understanding and readability by shortening and simplifying. It would be simpler to state Hacks law in deterministic form, state its inverse in deterministic form, and note that the goal is to derive its probabilistic representation from the theory of random walks in a Scheidegger network. The derivation of equation 7 as currently written, could then be simplified and disambiguated by

1. noting that width is the difference between two normally distributed and independent RVs and must be positive;

2. stating the ICs at $s = 0$, presumably as two rill sources separated by a distance $w = w_0$;

3. noting that the distribution of the difference of two independent normal distributions $z_1 - z_2$ with means $(\mu_1, \mu_2)$ and variances $(\sigma_1, \sigma_2)$ is itself a normal distribution with mean $\mu_1 - \mu_2$ and variance $\sigma_1^2 + \sigma_2^2$;

4. noting that the first passage of a random walk with a normally distributed RV with diffusion coefficient $D$ starting at $w = w_0$ is given by the Rayleigh distribution

$$f(w, s) = \frac{w}{4\pi D s^3} e^{-w^2/4Ds} \tag{1}$$

since, as written, the description is not clear, equation (5) is confusing, while the brief discussion of the diffusion (6) and Fokker-Planck equations will not be helpful to many readers.

**1.1 Typos, etc.**

1. Line 12 understanding Hack's

2. Line 34 geometrical (not topological)

3. Line 169

4. Line 197 the this

5. Line 286 , for ;

6. Line 330 length(en)

7. Line 343 geometrical (not topological)

8. Line 358 distribution a

Figure 12B needs some attention. Its not clear what its meant to be (as seen by this reviewer.)

**2 Comments Concerning Longer-Term Improvements**

The paper could be greatly improved if the authors are willing to put more work into both their numerical simulations and into their derivations, with the possibility of authoring two publishable papers. In terms of this option, a possible approach might be break the research into a subproject relating to each of the two main goals, with

1. the first subproject exploring the non-linearity of the relation in greater depth by running a large number of simulations over different experimental conditions and inferring the desired relations from the data;

2. the second subproject deriving Hack's law for Scheidegger networks with a deeper theoretical than is currently presented.

The first paper, for example, could describe the results of experiments that are run over a variety of rill geometries such as trapezoidal, triangular, and semi-circular etc. and over different erosion rules. This should led to interesting results, especially if variations in the results are related to different geomorphic variables.

The second paper could, for example, explore a somewhat more general approach to deriving a probability density function for the area-length relation by considering a a rilled slope of unbounded lateral extent (or some approximation) and considering an

ensemble of rills. A problem with the current approach of deriving this function from the difference of just two neighboring ridges is that the two adjacent rills that define a ridge at some point $s$ are not necessarily the ones that merge to close off the ridge since they might merge with their other adjacent rills. Intuitively this model would appear to lead to a different probability density function for area than the extended slope, with an increased probability of large and unrealistic widths. It is not clear, therefore, to this reviewer that this is the most appropriate model for deriving the Hack's law although its probably a good approximation. It is also possible that investigating an extended slope might lead to similar results to employing just two ridges or channels, but that would, of itself, be an interesting finding.

Another issue that could be explored in a second more theoretical paper is to work with a probablistic flow and transport theory to match the probablistic geometric theory. As currently written, the paper mixes probabilistic reasoning with deterministic reasoning in a somewhat simplistic manner. The geometrical modeling is probabilistic, while the modeling of the flow and erosion dynamics is deterministic. The problem with this approach is that the randomness in the whole system is driven entirely by the randomness in the geometry, which is not an entirely convincing assumption.

---

## Author Comment (AC1) · 30 Nov 2020

**1 Response to Reviewer 1**

We thank reviewer 1 for the thoughtful comments. We have considered many of their comments and applied we think that the updated manuscript addresses many of them.

The reviewer's main concern is with equation (2), from which many of the main results on  $f_A(A)$  follow. It is not clear how the authors obtained it and what assumptions are involved. In principle, the ensemble average of A, given I, is  $\langle A \rangle = \int A(f_l(l|A)f_A(A))/(f_l(l))dA$ . So, the ensemble average of A should contain information about the entire joint probability distribution. How does it reduce to equation (2)?

**Response**

Response: It is true that Hack's Law is a statement of the mean of the conditional distribution,

$$\mu_{(A|l)} = \int_{0}^{\infty} Af(A|l) \mathrm{d}A \,. \tag{1}$$

However, because  $f_l(l)$  doesn't appear anywhere in this expression, Hack's Law does not depend on the joint distribution - only on the conditional distribution. The joint distribution is important for our goal of understanding flow routing and distributions of hydraulic variables, but it is not necessary for understanding Hack's Law.

We suspect that the reviewer is questioning how the probabilistic representation of the mean of the conditional distribution  $f_A(A|l)$  yields Hack's law the specific form of Hack's Law written in (2). Hack's Law is an empirical scaling observation which we state in the beginning of this paragraph. Equation 2 is simply a restatement of (1) with *l* as the independent variable and so it is an empirical result. One of our goals is to demonstrate a reasoning for the nonlinear form of Hack's Law and the value of parameters  $\phi$  and *m*.

**ESurfD**
Is  $\phi$  really a constant?

**Response**

We now refer to  $\phi$  as a dimensional coefficient because, as the reviewer suggests,  $\phi$  can change depending on the class of network one is considering.

**1.0.1 Comment**

The reviewer suggests that we clarify the new elements of this work as compared to *Dodds and Rothman* (2000).

**1.1 Response**

We agree that much of this work builds on the reasoning from *Dodds and Rothman* (2000) and we have added language to more clearly acknowledge what components are new and which build on previous work. In particular, early on in section 2, we state that our primary contribution here is to contribute towards a formal understanding of the moments of the Hack distributions, whereas *Dodds and Rothman* (2000) developed the forms of the distributions.

**Comment**

The reviewer strongly suggests dividing section 2 and 3 in subsections.

**ESurfD**
**Response**

For section 2 we have divided into 3 sections: Geometry, Area, Length. For section 3 we have divided into 2 sections: Hydraulic distributions and Sampling.

**Minor Comments**

We have accepted the minor comments.

**2 Response to Reviewer 2**

We thank the reviewer for a careful reading and critique of our manuscript. The reviewer correctly points out some issues that may be altered to improve a revised draft. In our view, the reviewer's major contribution here highlights that we should clarify the reasoning for our initial and boundary conditions that determine the probability function for watershed width. Further they suggest that we clarify that our theory is based on a continuous random variable, whereas the Scheidegger model has discrete random variables.

**Comment**

Starting from Eq (5), this equation does not read as it meant "a unity probability with zero variance". It should be expressed as an atom of probability at w = 1 and zero for other w. Second, should it be w = 0 instead of w = 1? Right above this paragraph, it is stated that watersheds are closed at w = 0; thus they should also initiate at w = 0.
**Response**

We have changed language in the manuscript to highlight that the value of the Scheidegger network is to inform the probabilistic elements of constructing watersheds and the probability functions of geometric variables. Therefore, the discrete nature of the Scheidegger model is only to guide the mathematics for a continuous version. We agree with the reviewer that the initial condition should be represented as a dirac function and have made that change.

We disagree that the initial condition should be w = 0 for the following reason. It is true that at s = 0, the watershed has a width w = 0. However, in order for the watershed to exist, it necessarily must widen to a width of w = r at s = 1. This is where our initial condition applies and we have clarified this in the text

**Comment**

The reviewer has issue with our boundary conditions, which we state are fixed boundary conditions at w = 0. They suggest that given that it is possible for a watershed to close, that there is finite probability that w = 0 and therefore the boundary condition cannot be  $f_w(0) = 0$ .

**Response**

We understand the reviewers comment and how the previous manuscript led readers to that conclusion. However, we are confident in our boundary condition and have added language and a figure to explain why. Consider the ensemble of watersheds of length l. The width function, w(s) can take any random walk over the domain [0, l], but it must begin and end at w = 0. At s = 1 the variance of widths of the ensemble of watersheds of length l is 0. When s = l/2 the variance is at a maximum. While s > l/2 and as
 $s \rightarrow l$  the variance must begin to decline back towards 0. Therefore, the evolution of  $f_w(w,s)$  from 0 to l/2 is mirrored by the evolution from l/2 to l. Our boundary condition only applies to the  $s \leq l/2$ , or the 'growing' part of the watershed. The mirroring about l/2 then allows for us to know the form of the distribution over all s

The reviewer got the impression that at some point we were equating Scheidegger models and optimal criticality networks.

**Response**

We are uncertain about where the reviewer got this impression. We refer to OCNs in two places in the paper. The first instance, we say "Other network classes exist including optimal criticality networks..." We think that this clearly states that they differ from Scheidegger networks.

The other location that we refer to OCNs is in the discussion where we highlight the slight mismatch in form of the probability functions of area for real topography and a Scheidegger network. We have added language to make sure that readers do not think that we suggest Scheidegger and OCN are the same:

"However, because those networks are note amenable to the type of theory developed above because they lack the clarity in rules for links and nodes of the network. The Scheidegger model serves as a guide to inform probability distributions and provide a basic reasoning for nonlinear relationships."

Comment

Unclear how equation 11 is obtained, is it empirical?

**ESurfD**
**Response**

This result is semi-empirical. We expect that the variance of *A* scales as  $\sigma^2 l^3$ . The presence of 6 in the denominator is not theoretically derived. If one could formally explain this, then the problem would be complete. We have added:

"We emphasize that this is a semi-empirical result that warrants a stronger theoretical solution"

Comment

Rill flow length needs to be defined in the intro

Response

We have changed the appearance of "rill flow length" and now only have "watershed length" which is defined in the intro.

Comment

Clarify how rills are "efficient".

2.0.1 Response

We have removed the word "efficient"
Unclear: "second we ask if the particular arrangement of the rills focuses flow such that it leads to a nonlinear sediment yield relationship..."

**Response**

We have reworded this to: "Second, we ask if a well defined network of rills focuses flow such that it leads to a nonlinear sediment yield relationship"

Comment

OCN: Optimal Channel Networks?

Response

YES, we have changed.

Comment

Line 75- "As such, . . .": Is this referring to OCN? What is the "constraint"? OCN has a "clear rule" to construct networks.

Response

Yes it is referring to OCN, we think this is sufficiently clear. The constraint is that it minimizes the energy expenditure as stated above. We disagree that OCN has a clear rule
for constructing networks that leads to clear probabilistic insights. That OCN minimize energy and satisfy continuity equations does not lead to particularly clear conclusions regarding the construction of links and nodes. In contrast, the Scheidegger model is constructed by a set of uniformly spaced paths that take simple random walks in the cross slope dimension. Second, uniform drainage density is maintained so that when two rills meet, another is formed. These lead directly to a graphical representation of the network in a way that the rule for OCN does not. We do not think that further clarification benefits this manuscript as the focus is not on OCN.

**3 Reviewer 3**

We thank reviewer 3 for the meaningful review and suggestions on this manuscript. We believe that the manuscript is improved after following many of their suggestions.

**Comment**

... Since the primary goal is to demonstrate and understand the non-linearity of the length-sediment relation for a rilled surface, it would have been useful to have first discussed the relation for an unrilled surface...

**3.0.2 Response**

We agree with the reviewer and have added a paragraph dedicated towards addressing this. We review the work of Burch et al., 1986 which demonstrates nonlinear sediment yield-slope length relationship on planar surfaces. Their work suggests that the nonlinear relationship for planar and unrilled slopes approaches the least nonlinear relationship observed for rilled slopes. We suggest that this similarity is shared between

**ESurfD**
planar surfaces and rill networks with linear rills. We then review the work of *McGuire et al., 2013*, which demonstrates that rill networks become increasingly dendritic with increasing rainfall detachment. This is the scenario that leads to more nonlinear sediment yield relationships.

**Comment**

It would be useful to make a little section for related literature, and especially to add a comprehensive summary of all of the relevatn findings of Dodds and Rothman since they are employed/referred to so extensively.

**Response**

We have added a paragraph in the introduction summarizing these results and how we build on them.

**Comment**

The section on network geometry could be greatly improved for ease of understanding and readability by shortening and simplifying. It would be simpler to state Hacks law in deterministic form, state its inverse in deterministic form, and note that the goal is to derive its probabilistic representation from the theory of random walks in a Scheidegger network. The derivation of equation 7 as currently written, could then be simplified and disambiguated by

 noting that width is the difference between two normally distributed and independent RVs and must be positive;

**ESurfD**
- stating the ICs at s = 0, presumably as two rill sources separated by a distance w = w0;
- noting that the distribution of the difference of two independent normal distributions  $z_{-}z_{2}$  with means  $(\mu_{1},\mu_{2})$  and variances  $(\sigma_{1},\sigma_{2})$  is itself a normal distribution with mean  $\mu_{1} - \mu_{2}$  and variance  $\sigma_{1} + \sigma_{2}$ ;
- noting that the first passage of a random walk with a normally distributed RV with diffusion coefficient D starting at w = w0 is given by the Rayleigh distribution

since, as written, the description is not clear, equation (5) is confusing, while the brief discussion of the diffusion (6) and Fokker-Planck equations will not be helpful to many readers.

**Response**

We have largely taken these suggestions. We have clarified the initial conditions and added a figure that explains them and the boundary conditions. In our case, the Rayleigh distribution represents the distribution of watershed widths for  $s \leq l/2$ , which we have clarified in this case. We disagree though that w(s) is a first passage problem because a watershed can return to the same value for w at many points along the watershed. We have; however, chosen to remove most references to the Fokker-Planck equation for simplicity.

**Minor Comments**

- 1. Line 12 understanding Hack's
- 2. Line 34 geometrical (not topological)
- 3. Line 169

**ESurfD**
4. Line 197 the this
5. Line 286 , for ;
6. Line 330 length(en)
7. Line 343 geometrical (not topological)
8. Line 358 distribution a

Figure 12B needs some attention. Its not clear what its meant to be (as seen by this reviewer.)

**Response**

We have accepted most of the minor comments that identify typos or improved language. Figure 12B is included to illustrate the results from flow routing. We have added the following language:

"This illustrates the results of numerical flow routing. From this result, we calculate exceedance probabilities that compare to theoretical distributions."

Response to longer term improvements

We appreciate the reviewers suggestion regarding future developments and their support for further work. Indeed some form of what they mention is the subject of current work. We choose to keep the focus of this manuscript largely the same as in the previous manuscript. However, we respond to a few points here.

First, we address their suggestions for a second paper and uncertainty with regard to a Scheidegger model being suitable for understanding Hack's Law. We understand that the reviewer thinks that the Scheidegger model is likely a good model, but not necessarily the best because two adjacent rills that define a ridge may not be the ones Interactive comment

that meet to close it off. This is true, however, the observation that watershed divides take simple random walks and necessarily begin at w = 1 at s = 1 holds. In fact, the ridges form a network that is also a Scheidegger network and is simply the complement of the channels. We think that insofar as the Scheidegger model is a simplification of networks, it is a reasonable one for exploring Hack distributions and laws.

Second, we agree with the reviewer's comment that our representation of channel widths is somewhat simplistic. One could reasonably make  $r_w$  a random variable that scales with Q and we think this would be a fruitful next step, but beyond the scope of this paper.

**ESurfD**

**ESurfD**

---

## Author Response (AR2)

Author Response:

We thank the editor, Dr. Passalacqua, for doing such a careful read of the manuscript and catching all of the errors and making suggestions. I have accepted all of the suggested edits and made some additional edits to the text that clarify the writing. New wording is highlighted in red in this version. All edits are for clarity and do not change content.

One notable change that I've made is exchanging references of the flux at the base of a hillslope, $q$ [$L^2 T^{-1}$] to $Q_s$ [$L^3 T^{-1}$] which we refer to as a volumetric sediment yield. This changes nothing about the theory, figures, or results, we just found that it is simpler to consider when we talk about sediment yield. Again, this edit is for clarity and changes nothing about the concepts.

Here are the minor comments made by the editor, Dr. Paola Passalacqua:

The main comments have been addressed but several parts of the text remain rough, particularly the new ones, but there are also numerous typos remaining after the previous revisions. Please make sure you address the edits below but also carefully read and edit as needed the manuscript throughout as I may have missed some. The number indicates the line.

8 – 'relationship'
Changed

17 - 'serve as'
Changed

21 – 'hillslope'
Reworded this sentence

27 - 'that are of the form' or 'a nonlinear relationship'
Changed

29 - focused with distance does not sound right:
Changed wording

46-48 - you could use this last sentence to express more clearly what knowledge gaps remained from those previous works:
Rephrased this sentence to discuss the underlying probability functions that drive sediment yield

62 - 'Addressing these goals involves'
Changed

91 - ' that model is unrealistic'
Changed

104 - this seems already stated above
Removed

118 - 'useful and simplified'
Changed

144 - 'a finite'
Changed

148 – 'know'
Changed wording of this sentence

158 - 'of the'
Changed

173 – 'increases'
Changed

175 - 'we obtain'
Changed

228 – 'detachment,'
Changed

390 - 'for the specific form adopted here' or 'for the value of six in the denominator'
Changed

394 - 'essence of'
Changed

409 - incomplete sentence
Holy cow. Yes. Changed.

411 - sentence needs to be revised
Oof. Yes, We have changed this.

447 - missing subject
added 'we have'